# Utility of 3T MRI in Women with IB1 Cervical Cancer in Determining the Necessity of Less Invasive Surgery

**DOI:** 10.3390/cancers14010224

**Published:** 2022-01-04

**Authors:** Soo Young Jeong, Byung Kwan Park, Chel Hun Choi, Yoo-Young Lee, Tae-Joong Kim, Jeong-Won Lee, Byoungi-Gie Kim

**Affiliations:** 1Hallym University Medical Center, Department of Obstetrics & Gynecology, Kangnam Sacred-Heart Hospital, Hallym University College of Medicine, Seoul 07441, Korea; ohora_87@naver.com; 2Samsung Medical Center, Department of Radiology, Sungkyunkwan University School of Medicine, Seoul 06351, Korea; 3Samsung Medical Center, Department of Obstetrics & Gynecology, School of Medicine, Sungkyunkwan University, Seoul 06351, Korea; yyl.lee@samsung.com (Y.-Y.L.); tj28.kim@samsung.com (T.-J.K.); garden.lee@samsung.com (J.-W.L.); bksong.kim@samsung.com (B.-G.K.)

**Keywords:** high tesla magnetic resonance imaging, early cervical cancer, less invasive surgery

## Abstract

**Simple Summary:**

3T MRI can estimate more precisely the tumor volume of early cervical cancer than physical examination. Women with IB1 cervical cancer, which is invisible on 3T MRI, have no parametrial invasion so that parametrectomy can be skipped or minimized. Vagina invasion or lymph node metastasis is rare in these women so that vaginectomy or lymph node dissection can be performed less aggressively. Therefore, less invasive surgery can be one of the treatment options if IB1 cervical cancer is invisible on 3T MRI.

**Abstract:**

Purpose: Cervical cancer that is invisible on magnetic resonance imaging (MRI) may suggest lower tumor burden than physical examination. Recently, 3 tesla (3T) MRI has been widely used prior to surgery because of its higher resolution than 1.5T MRI. The aim was to retrospectively evaluate the utility of 3T MRI in women with early cervical cancer in determining the necessity of less invasive surgery. Materials and methods: Between January 2010 and December 2015, a total of 342 women with FIGO stage IB1 cervical cancer underwent 3T MRI prior to radical hysterectomy, vaginectomy, and lymph node dissection. These patients were classified into cancer-invisible (*n =* 105) and cancer-visible (*n =* 237) groups based on the 3T MRI findings. These groups were compared regarding pathologic parameters and long-term survival rates. Results: The cancer sizes of the cancer-invisible versus cancer-visible groups were 11.5 ± 12.2 mm versus 30.1 ± 16.2 mm, respectively (*p* < 0.001). The depths of stromal invasion in these groups were 20.5 ± 23.6% versus 63.5 ± 31.2%, respectively (*p* < 0.001). Parametrial invasion was 0% (0/105) in the cancer-invisible group and 21.5% (51/237) in the cancer-visible group (odds ratio = 58.3, *p* < 0.001). Lymph node metastasis and lymphovascular space invasion were 5.9% (6/105) versus 26.6% (63/237) (5.8, *p* < 0.001) and 11.7% (12/105) versus 40.1% (95/237) (5.1, *p* < 0.001), respectively. Recurrence-free and overall 5-year survival rates were 99.0% (104/105) versus 76.8% (182/237) (*p* < 0.001) and 98.1% (103/105) versus 87.8% (208/237) (*p* = 0.003), respectively. Conclusions: 3T MRI can play a great role in determining the necessity of parametrectomy in women with IB1 cervical cancer. Therefore, invisible cervical cancer on 3T MRI will be a good indicator for less invasive surgery.

## 1. Introduction

Radical hysterectomy, vaginectomy, and lymph node (LN) dissection have been considered as the standard treatment in treating International Federation of Gynecology and Obstetrics (FIGO) stage IB1 cervical carcinoma. Many investigations have demonstrated that these surgical procedures provide good long-term survival rates in women with this early cervical cancer [1,2,3,4].

However, these surgical procedures also may induce various complications. Bladder anatomy is gradually deformed, and the bladder function becomes poor because radical hysterectomy is associated with parametrectomy, leading to autonomic nerve injury [2,3,4]. Furthermore, this nerve injury may cause anorectal motility disorder and sexual dissatisfaction [5,6,7]. If the vaginectomy becomes excessive, women with early cervical cancer cannot feel sexual satisfaction postoperatively. Lymph node dissection may lead to lymphedema in women with IB1 cervical cancer [8,9,10]. Higher availability of screening examination helps to detect early cervical cancer in relatively young women. Subsequently, they have to face the poor quality of life resulting from life-long postoperative morbidities.

Magnetic resonance imaging (MRI) is more precise in estimating tumor volume than is physical examination because MRI enables accurate measurement of three-dimensional tumor axes [11,12,13]. Only a few investigations have reported on the usefulness of such MRI findings, showing that early cervical or endometrial cancer can be treated with less invasive surgery if the tumor is invisible on MRI [14,15,16]. However, these studies did not deal with the role of 3 tesla (3T) MRI in evaluating early cervical cancer. 3T MRI provides a higher image resolution or shorter scan time compared to 1.5T MRI [17,18,19]. Therefore, we hypothesized that 3T MRI can provide useful imaging findings to determine whether less invasive surgery is necessary.

The aim of this study was to retrospectively evaluate the utility of 3T MRI in women with early cervical cancer in determining the necessity of less invasive surgery.

## 2. Materials and Methods

This study (File No.: 2018-06-114) was approved by our institutional review board in Samsung Medical Center and informed consent was waived due to the retrospective design.

### 2.1. Patients

Between January 2010 and December 2015, a total of 427 patients with FIGO IB1 cervical cancer underwent MRI prior to radical hysterectomy (Figure 1). Among them, 85 patients were excluded due to MRI examinations that were scanned using a 1.5T scanner or were done in a local hospital. Finally, 342 patients were included in the study population when they underwent 3.0T MRI at a single institute. Of them, 105 women (cancer-invisible group) had a cancer that was invisible on MRI. The remaining 237 women (cancer-visible group) had a cancer that was visible on MR images. The medical records of the cancer-invisible group (age range, 27–81 years; mean ± standard deviation, 48.1 ± 11.4 years) and cancer-visible group (25–81 years; 50.1 ± 11.5 years) were reviewed. Colposcopic biopsy and conization were performed in 70.8% (242/342) and 29.2% (100/342), respectively.

Bimanual pelvic and rectovaginal examinations were done to identify the disease extent. Laboratory tests, chest radiography, cystoscopy, and sigmoidoscopy were routinely performed for the clinical FIGO staging [1]. The time interval between MRI and hysterectomy ranged from 1 to 115 days (15.4 ± 12.8 days) in the cancer-invisible group and from 0 to 79 days (14.3 ± 9.3 days) in the cancer-visible group.

The MR images were preoperatively interpreted by one of two radiologists who had approximately 5 years of experience in gynecologic imaging and were additionally reviewed by one radiologist who had approximately 19 years of experience in gynecologic imaging. The MRI diagnoses of only three cases were changed from the cancer-invisible group to cancer-visible group.

Radical hysterectomy, vaginectomy, and LN dissection were performed in all women. Additional surgical procedures depended on the clinical stage and the surgeons’ decision. When pelvic lymph nodes were suspicious for metastasis at frozen sectioning, para-aortic LNs were dissected.

Two pathologists examined radical hysterectomy, vaginectomy, and LN specimens. They recorded the size of the cervical cancer, histologic type, depth of stromal invasion, lymphovascular space (LVS) invasion, parametrial invasion, vaginal invasion, resection tumor margin, and LN metastasis.

After primary treatment, all patients received adequate follow-up procedures. During this period, patients underwent physical examination, Pap smear, and tumor marker every 3 months for the first 2 years, and every 6 months for the next 3 years. Imaging studies, such as abdomiopelvic CT or pelvis MRI, were conducted every 6–12 months for the first 2 years and then annually for the next 3 years.

### 2.2. MR Imaging

The pelvis was scanned with a 3T MRI scanner (Intera Achiva 3T; Philips Medical System, Best, The Netherlands). The upper abdomen was scanned with a 3T MRI or CT. MRI sequences of the pelvis included T2-weighted images, T1-weighted images, diffusion-weighted images, and dynamic contrast-enhanced images. T2-weighted images were obtained into axial, sagittal, and coronal planes. The other sequences were obtained in the axial planes. The upper abdomen was scanned from lower lung to aortic bifurcation. T2-weighted and T1-weighted axial images were obtained using a fast spin echo sequence. We used the same MR parameters as those that Park et al. used [2].

### 2.3. Data Analysis

Invisible cancer was defined when the cervical tumor was not seen on either the T2-weighted images, diffusion-weighted images, or contrast-enhanced T1-weighted images (Figure 2) [3]. Visible cancer was defined when the cervical cancer is hyperintense on T2-weighted images and hyperintense on diffusion-weighted images, hypointense on apparent diffusion coefficient map images, and poorly enhanced on contrast-enhanced T1-weighted images, as compared to neighboring cervical tissue (Figure 3) [3]. Post-biopsy inflammation was differentiated from cervical cancer with the following findings: it was hyperintense on T2-weighted images. However, it had no diffusion restriction and showed iso- or higher enhancement compared to neighboring cervical tissue on post-contrast MR images [3].

Cancer-invisible and cancer-visible groups were compared regarding patient age, biopsy type, histologic type, and squamous cell carcinoma (SCC) antigen. These groups were also compared in terms of residual tumor size, depth of stromal invasion, LVS invasion, parametrial invasion, vaginal invasion, and LN metastasis. Post-biopsy tumor sizes on MR images were correlated with those on radical hysterectomy.

Recurrent tumor was assessed on the follow-up CT or MR images. Recurrence-free and overall 5-year survival rates were calculated. Cancer-invisible and cancer-visible groups were compared regarding the recurrent rate and recurrence-free or overall 5-year survival rate.

### 2.4. Statistical Analysis

Patient age, tumor size, SCC antigen, and invasion depth were compared with a Mann–Whitney test because these data had a non-Gaussian distribution.

Proportions of biopsy types, cancer histology, LVS invasion, parametrial invasion, vaginal invasion, LN metastasis, and recurrent rate were compared with Fisher’s exact test.

Odds ratios and 95% confidence intervals were calculated using the approximation of Woolf. When a value was zero, 0.5 was added to each to make the calculation possible. 

Recurrence-free and overall 5-year survival rates were compared with Kaplan–Meier survival curves.

Commercially available software SPSS 24.0 for Windows (SPSS Inc., Chicago, IL, USA) was used for statistical analyses. A *p*-value of <0.05 was considered statistically significant.

## 3. Results

The cancer-invisible group underwent conization in 59.0% (62/105) and coloposcopic biopsy in 41.0% (43/105), while the cancer-visible group did conization in 16.0% (38/237) and colposcopic biopsy in 84.0% (199/237), respectively (*p* < 0.001) (Table 1). There was a significant difference between the groups in terms of histologic types (*p* = 0.046). The cancer-invisible group had a lower proportion of SCC (*p* = 0.039) and a higher proportion of adenocarcinoma than the cancer-visible group (*p* = 0.015) (Table 1). The cancer-invisible group had a higher level of SCC antigen than the cancer-visible group.

The median residual tumor size was 11.5 mm (0–55.0 mm) in the cancer-invisible group and 30.1 mm (0–95.0 mm) in the cancer-visible group (*p* < 0.001) (Table 2). The median depth of stromal invasion was 20.5% (0–100%) in the cancer-invisible group and 63.5% (0–100%) in the cancer-visible group (*p* < 0.001). Residual tumors of these groups were detected in 67.6% (71/105) and in 97.0% (230/237) (*p* < 0.001), respectively. Accordingly, the cancer-invisible group had no residual tumor more frequently than the cancer-visible group (*p* < 0.001). Parametrial invasion in the cancer-invisible and cancer-visible groups was detected in 0% (0/105) and 21.5% (51/237), respectively (*p* < 0.001). LVS invasion was 11.7% (12/105) in the cancer-invisible group and 40.1% (95/237) in the cancer-visible group, respectively (*p* < 0.001). LN metastasis of these groups was detected in 5.9% (6/105) and 26.6% (63/237), respectively (*p* < 0.001). However, vaginal invasion of these groups was detected in 1.9% (2/105) and 5.5% (13/237), respectively (*p* = 0.163).

The tumor recurrent rate was 1.0% (1/105) in the cancer-invisible group and 23.2% (55/237) in the cancer-visible group on follow-up CT or MR images (*p* < 0.001). The recurrence-free survival rates of the cancer-invisible and cancer-visible groups were 99.0% (104/105) and 76.8% (182/237) (*p* < 0.001), respectively (Figure 4A). The overall 5-year survival rates of these groups were 98.1% (103/105) and 87.8% (208/237) (*p* = 0.003), respectively (Figure 4B).

Among the post-hysterectomy histologic findings, parametrial invasion provided the highest odds ratio, 53.8 in the cancer-visible group versus the cancer-invisible group (Table 3). The other odds ratios were 31.4, 15.7, 5.8, 5.1, and 0.3 regarding recurrent tumor, residual tumor, LN metastasis, LVS invasion, and vaginal invasion, respectively.

## 4. Discussion

Our study showed that the cancer-invisible group had no parametrial invasion postoperatively. Furthermore, the residual tumor and invasion depth were much smaller than those in the cancer-visible group. The incidences of LN metastasis and LVS invasion were also much lower than those in the cancer-invisible group. Subsequently, the cancer-invisible group had a much higher long-term recurrence-free or overall survival rate than the cancer-visible group. 

Parametrectomy is performed for radical hysterectomy in women with early cervical cancer to reduce recurrent cervical cancer [4,5,6,7]. However, several investigations have reported that a small tumor size, small depth of invasion, or no LVS invasion are good prognostic factors for no parametrial involvement [8,9,10,11,12,13]. The risk factor that they predict parametrial invasion most commonly is the tumor size [8,9,10,11,12,13]. If it is 2 cm or less, parametrial invasion is rare so that less invasive surgery is recommended in this small cervical cancer [8,9,10,11,13]. Therefore, parametrectomy appears aggressive in these clinical settings.

From this point of view, invisible cancer on 3T MRI can be a much stronger indicator than small tumor size. Our study showed that parametrial invasion was postoperatively absent in the cancer-invisible group. Moreover, the odds ratio of this histologic finding was much higher compared to those of the other histologic findings. Accordingly, invisible cancer on 3T MRI can strongly suggest no sign of parametrial invasion [3]. Park et al. also have reported that there was no parametrial invasion in patients with 1B1 cervical cancer that was invisible on 1.5T or 3T MRI [2]. However, the number of patients undergoing 3T MRI was small in their study. Kamimori and Yamajaki et al. also have reported that parametrial invasion is rare in small cervical cancer preoperatively measured on preoperative MRI [12,13]. They did not state whether their MRI scanner is 1.5T or 3T. 

The other prognostic factors are still difficult to precisely identify with preoperative MRI. LN metastasis was most commonly assessed with MRI. Previously reported papers have showed that MRI sensitivity for detecting LN metastasis was only 30–73% [14,15,16,17]. They used the lymph node size in order to determine if there was metastasis. Diffusion-weighted MRI improves the diagnostic accuracy, but the sensitivity and specificity were 86% and 84%, respectively [18]. Our study also showed that if IB1 cancer was invisible on 3T MRI, then the incidence of LN metastasis was much smaller than that in their study [18] with diffusion-weighted imaging. However, further investigation is necessary to determine if LN dissection can be skipped in invisible IB1 cervical cancer. The incidence of LN metastasis was slightly higher in the cancer-invisible group compared to that in the Park et al. investigation [2]. LVS invasion or depth of stromal invasion is still impossible to detect with preoperative MRI. After all, assessing tumor depiction on 3T MRI is a good prognostic factor for predicting parametrial invasion in patients with 1B1 cervical cancer. Ultrahigh-field MRI at 7T or higher will be introduced in the near future and these prognostic factors can then be assessed more precisely [19,20].

Parametrectomy may injure the ureter or nerve in the parametrium [5,6,7,21]. Ureter injury manifests postoperatively in urine leakage or ureter obstruction [21]. This complication needs interventional or surgical procedures. Furthermore, many bundles of autonomic nerves are interrupted by parametrectomy so that autonomic function of the bladder, vagina, or rectum can be impaired. Subsequently, follow-up CT and MRI can show gradual deformity of the urinary bladder, such as a thick wall, coarse trabeculation, residual urine, or over-distended bladder [5,6,7]. Neurogenic bladder is not uncommon after radical hysterectomy, as many patients undergo urinary frequency changes, recurrent cystitis, or self-catheterization. The autonomic nerve injury may also induce sexual or anorectal dysfunction [22,23,24]. Therefore, parametrectomy should be avoided in patients who do not have parametrial invasion.

The incidence of invisible 1B1 cancer on MRI is not well-known. Park et al. have reported that it accounts for 24.9% (86/346) among the IB1 cervical cancers [2]. Our study showed that it slightly increased to 30.7% (105/342). More available screening tests and MRI examinations may increase the early detection of IB1 cervical cancer [3]. Besides, the ongoing development of MRI techniques will provide more precise information on cancer detection. Importantly, most of those patients have early stage disease. As a result, parametrectomy can be skipped in a larger number of patients with 1B1 cervical cancer. The incidence of postoperative complications will be reduced with less invasive surgery. Nevertheless, gadolinium has been found to cross the placenta and to stimulate malformations in animal models [25]. Hence, its use during pregnancy is contraindicated in the first trimester of pregnancy in patients with cervical cancer and improved MRI techniques are warranted [26].

Compared to the results of Park et al.’s study, the incidences of residual tumor, vaginal invasion, lymph node metastasis, and lymphovascular space invasion were slightly higher in our study. Our study showed that the proportion (38.1%) of adenocarcinoma was relatively higher compared to that (29.1%) in their study. Thus, we think that increasing the proportion of non-SCC cervical cancer might influence the incidence differences of other pathologic findings. An invisible tumor on 3T MRI does not contribute to skipping vaginectomy and lymph node dissection in patients with 1B1 non-SCC cervical cancer. Further investigations are necessary to compare SCC and non-SCC in terms of MRI findings or clinical outcomes.

This study has some limitations. First, it was conducted using a retrospective design. The likelihood for selection bias cannot be excluded. This limitation may influence the histologic type of cervical cancers. Second, baseline characteristics were not matched with propensity scores between the groups. Third, the incidence of non-squamous cell cancers was relatively higher than that of previously published studies. This finding might result from selection bias in that many squamous cell carcinomas unfit given the inclusion criteria were excluded. These cancers showed poor behavior compared to SCC. This finding might influence the residual tumor, vaginal invasion, LVS invasion, or LN metastasis. Fourth, post-operative complications were not qualitatively or quantitatively assessed to compare between the groups. We relied on follow-up CT or MRI findings to detect anatomical changes. However, this assessment is not sufficient to precisely identify functional changes. Fifth, our scanners were the initial version of 3T MRI. Currently these scanners are replaced with upgraded 3T MRI. Furthermore, non-SCC cervical cancer had a greater number of residual tumors than SCC cervical cancer. These factors may result in discordant findings between 3T MRI and pathologic examination in terms of residual tumors. Advanced ultrasonography with 3D and color Doppler display as well as transvaginal elastography also may be useful to evaluate early cervical cancer [27].

## 5. Conclusions

3T MRI can be a useful application to guide surgical interventions in patients with IB1 cervical cancer. If a cervical cancer is not depicted on 3T MRI, gynecologists can skip parametrectomy or remove as little as possible of the parametrial tissue. Subsequently, postoperative complications, such as bladder dysfunction, sexual dissatisfaction, or anorectal dysfunction, will be reduced in patients with invisible IB1 on 3T MRI. However, invisible cancer on 3T MRI cannot completely exclude the likelihood of vaginal invasion or LN metastasis, even if the incidences are much lower than those in patients with visible IB1 cervical cancer on MRI. Future introduction of higher than 3T MRI will contribute to better determining if less invasive surgery is necessary.

## Figures and Tables

**Figure 1 cancers-14-00224-f001:**
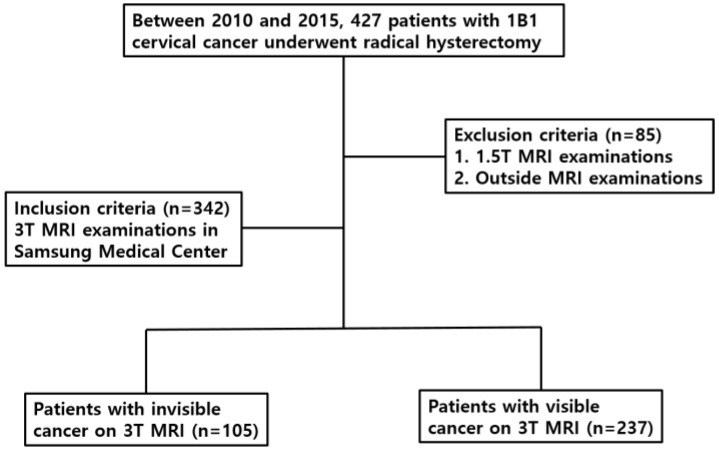
Flow diagram of selecting the study population.

**Figure 2 cancers-14-00224-f002:**
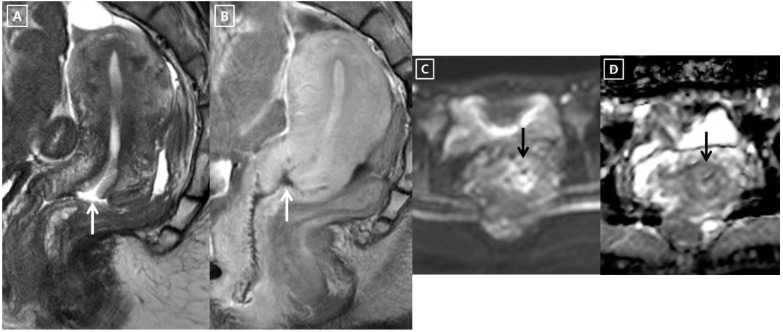
A 41-year-old woman with invisible IB1 cervical cancer on 3T MRI. (**A**,**B**). T2-weighted and delayed contrast-enhanced sagittal MR images do not show residual cancer but a conization defect (white arrow). The radical hysterectomy specimen demonstrated a 2.0 mm residual SCC and 22.2% invasion depth. There was no invasion or metastasis into the parametrium, vagina, LVS, and LN. (**C**,**D**). The diffusion-weighted axial MR image shows a hyperintense cervix (black arrow). However, it does not show a low ADC value (black arrow) on the apparent diffusion coefficient map image. Therefore, cervical cancer is not visible on the MR images.

**Figure 3 cancers-14-00224-f003:**
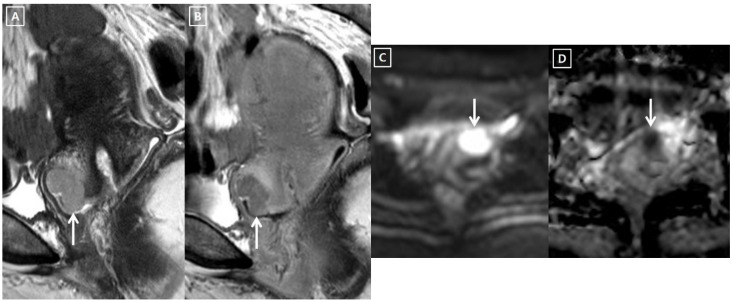
A 40-year-old woman with visible IB1 cervical cancer on 3T MRI. (**A**,**B**). T2-weighted sagittal MR image following colposcopic biopsy show a slightly hyperintense cervical cancer. This tumor shows poor enhancement on delayed contrast-enhanced sagittal MR image. (**C**,**D**). The diffusion-weighted axial MR image shows a hyperintense cervical mass (white arrow), suggesting strong diffusion restriction. The tumor (white arrow) is hypointense on an apparent diffusion coefficient axial MR image. The radical hysterectomy specimen showed a 20 mm residual cancer (Glassy cell carcinoma) and 40.0% invasion depth. Four metastatic lymph nodes were detected even though there was no invasion into the parametrium, vagina, and LVS.

**Figure 4 cancers-14-00224-f004:**
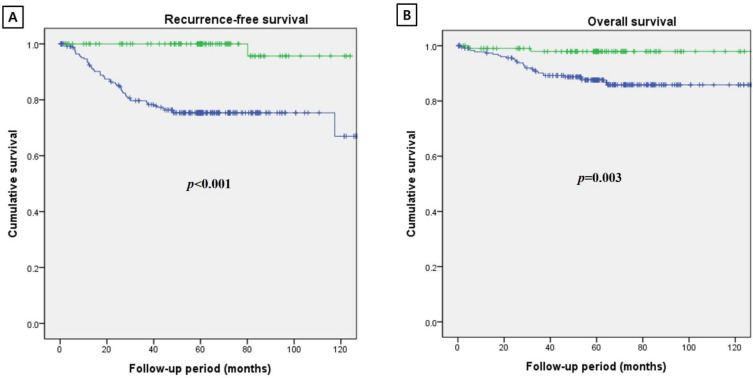
Kaplan–Meier survival curves in patients with IB1 cervical cancer. (**A**) The cancer-invisible group (green line) achieves a higher recurrence-free 5-year survival rate (99.0% versus 76.8%) than the cancer-visible group (blue line) (*p* < 0.001). (**B**) The cancer-invisible group (green line) achieves a higher overall 5-year survival rate (98.1% versus 87.8%) than the cancer-visible group (blue line) (*p* = 0.003).

**Table 1 cancers-14-00224-t001:** Comparison of patient demographics between the groups.

Demographics	Patients (*n =* 342) with FIGO Stage IB1 Cervical Cancer: Cancer Visibility on 3T MRI		*p* Value
	Cancer-Invisible Group (*n =* 105)	Cancer-Visible Group (*n =* 237)	
Age (years)	48.1 ± 11.4 (27–81)	50.1 ± 11.0 (25–81)	0.073
Conization	62 (59.0%)	38 (16.0%)	<0.001
Colposcopic biopsy	43 (41.0%)	199 (84.0%)	<0.001
SCC	58 (55.2%)	159 (67.1%)	0.039
Adenocarcinoma	40 (38.1%)	59 (24.9%)	0.015
Other cancers	7 (6.7%)	19 (8.0%)	0.326
SCC antigen (ng/mL)	1.4 ± 3.0 (0.2–21.4)	4.6 ± 12.2 (0.1–91.7)	<0.001

Note—SCC, squamous cell carcinoma. Age and SCC antigen are shown as the median ± standard deviation (range).

**Table 2 cancers-14-00224-t002:** Comparison of post-hysterectomy findings between the groups.

Histologic Findings	Patients (*n =* 342) with FIGO Stage IB1 Cervical Cancer: Cancer Visibility on 3T MRI		*p* Value
	Cancer-Invisible Group (*n =* 105)	Cancer-Visible Group (*n =* 237)	
Tumor size (mm)	11.5 ± 12.2 (0–55.0)	30.1 ± 16.2 (0–95.0)	<0.001
Invasion depth (%)	20.5 ± 23.6 (0–100)	63.5 ± 31.2 (0–100)	<0.001
Residual tumor	71 (67.6%)	230 (97.0%)	<0.001
LVS invasion	12 (11.7%)	95 (40.1%)	<0.001
Parametrial invasion	0 (0.0%)	51 (21.5%)	<0.001
Vaginal invasion	2 (1.9%)	13 (5.5%)	0.163
LN metastasis	6 (5.9%)	63 (26.6%)	<0.001

Note—LN, lymph node; LVS, lymphovascular space. Tumor size and invasion depth are shown as the median ± standard deviation (range).

**Table 3 cancers-14-00224-t003:** Odd ratios of the post-hysterectomy and imaging findings.

Histologic and Imaging Findings	Cancer-Visible Group versus Cancer-Invisible Group		*p* Value
	Odds Ratio	95% Confidence Interval	
Parametrial invasion	58.3	3.6–956.5	<0.001
Recurrent tumor	31.4	4.3–230.5	<0.001
Residual tumor	15.7	6.7–37.0	<0.001
LN metastasis	5.8	2.4–13.9	<0.001
LVS invasion	5.1	2.6–9.8	<0.001
Vaginal invasion	0.3	0.07–1.51	0.163

Note—LN, lymph node; LVS, lymphovascular space.

## Data Availability

The data presented in this study are available on request from the corresponding author.

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
