# Peer review of "Utility of 3T MRI in Women with IB1 Cervical Cancer in Determining the Necessity of Less Invasive Surgery"

_cancers, 2022, doi:10.3390/cancers14010224_

Round 1
Reviewer 1 Report
General Comments:
It is hard to believe that 3T MRI missed tumors with 1.1 cm size since the spatial resolution is 1 mm. Other studies using 1.5T MRI (Reference 11, 21-24) or ultrasonography reported similar conclusions (Gyn Oncol 2013; 128:449. Ultrasound Ob Gyn 2015; 45:459). It is better to make a comment on advanced ultrasonography with 3D and color Doppler display as well as transvaginal elastography to evaluate early cervical cancer.
SPECIFIC COMMENTS:
- Fig. 1: What size of the tumor was invisible?
- Line 95: .....LVS------>.....lymphovascular space (LVS)
- line 180: ......common------->.....commonly
- line 233:..... high------->.....higher
- Results: There is no page number. ....and------>....and
- Why do your study show higher lymph node metastasis in invisible cancers when compared with Reference 14?
- Why do your study show high proportion of adenocarcinoma in invisible cancer group? The sentence on line 226-228 should be clarified for the clear message.
Author Response
Authors responses to reviewer #1 comments:
It is hard to believe that 3T MRI missed tumors with 1.1 cm size since the spatial resolution is 1 mm. Other studies using 1.5T MRI (Reference 11, 21-24) or ultrasonography reported similar conclusions (Gyn Oncol 2013; 128:449. Ultrasound Ob Gyn 2015; 45:459). It is better to make a comment on advanced ultrasonography with 3D and color Doppler display as well as transvaginal elastography to evaluate early cervical cancer.
Response: Thank you for your comments. Our MRI scanners were the initial version of 3T MRI. Now these scanners are replaced with upgraded 3T MRI. Besides, non-SCC cervical cancer had a greater number of residual tumor than SCC cervical cancer. These factors including the types of scanner and histology may result in discordant findings between 3T MRI and pathologic examination in terms of residual tumor. Advanced ultrasonography with 3D and color Doppler display as well as transvaginal elastography may be useful to evaluate early cervical cancer. It will be added as one of limitations.
SPECIFIC COMMENTS:
- Fig. 1: What size of the tumor was invisible?
Response: Conization biopsy reported that the tumor size was 0.9 cm. Radical hysterectomy showed that the residual tumor was 2 mm in horizontal spread.
- Line 95: .....LVS------>.....lymphovascular space (LVS).
Response: Thank you for your comment. We will rephrase it into “lymphovascular space (LVS)”.
- line 180: ......common------->.....commonly
Response: Thank you for your comment. We will rephrase it into “commonly”.
- line 233:..... high------->.....higher
Response: Thank you for your comment. We will rephrase it into “higher”.
- Results: There is no page number. ....and------>....and
Response: Thank you for your comment. We will rephrase it into “and”.
- Why do your study show higher lymph node metastasis in invisible cancers when compared with Reference 14?
Response: Thank you for your comment. We think that the proportions of non-SCCs was significantly higher than that in reference 14 study. The histologic type of cervical cancer might influence on lymph node metastasis.
- Why do your study show high proportion of adenocarcinoma in invisible cancer group? The sentence on line 226-228 should be clarified for the clear message.
Response: Thank you for your comment. As we stated in the limitations, the design of our study was conducted retrospectively. Therefore, we cannot completely selection bias that many SCC, which were unmet for inclusion criteria, were excluded. We will add these statement in the limitations.
Reviewer 2 Report
In this article, the authors aimed to evaluate the utility of 3T MRI in women with early cervical cancer in determining the necessity of less invasive surgery. The manuscript is straightforward, well written, and concise and has clear results within the scope of a retrospective study. Definitely deserves to be published and is a valuable contribution to the “cancers” journal. Some minor flaws need to be addressed before publication.
Minor points:
[1] “Discussion”, Page 8 of 11, Lines 219-220:
“Besides, ongoing development of MRI techniques will provide more precise information on cancer detection.”.
The ongoing development of the MRI techniques may have multiple benefits. Importantly, cervical cancer is the most commonly diagnosed gestational gynaecological malignancy and most of those patients have early stage disease. MRI is useful for evaluation of masses that are difficult to visualize with ultrasound; nevertheless, gadolinium has been found to cross the placenta and to stimulate malformations in animal models. Hence, its use during pregnancy is contraindicated in the first trimester of pregnancy and improved MRI techniques are warranted.
Recommended reference: Boussios S, et al. A review on pregnancy complicated by ovarian epithelial and non-epithelial malignant tumors: Diagnostic and therapeutic perspectives. J Adv Res. 2018;12:1-9.
[2] “Discussion”, Page 8 of 11, Lines 226-228:
“Thus, we think that increasing proportion of non-SCC cervical cancer might influence the incidence differences of other pathologic findings because it provides worse prognosis than SCC cervical cancer”.
At that point, the authors should clarify that the difference in survival outcomes of squamous cell carcinomas and adenocarcinomas of the uterine cervix is still controversial. It has not been elucidated whether patients with cervical adenocarcinomas have a higher incidence of pelvic lymph node involvement compared to those with squamous cell carcinomas. Probably, the worse outcome of patients with adenocarcinomas is due to ineffective adjuvant treatment, rather than a higher incidence of lymph node involvement as compared to squamous cell carcinomas.
[3] General comment:
A workflow diagram for the study would be of benefit for the readers.
Author Response
Authors responses to reviewer #2 comments:
In this article, the authors aimed to evaluate the utility of 3T MRI in women with early cervical cancer in determining the necessity of less invasive surgery. The manuscript is straightforward, well written, and concise and has clear results within the scope of a retrospective study. Definitely deserves to be published and is a valuable contribution to the “cancers” journal. Some minor flaws need to be addressed before publication.
Response: Thank you for your comments.
Minor points:
[1] “Discussion”, Page 8 of 11, Lines 219-220:
“Besides, ongoing development of MRI techniques will provide more precise information on cancer detection.”.
The ongoing development of the MRI techniques may have multiple benefits. Importantly, cervical cancer is the most commonly diagnosed gestational gynaecological malignancy and most of those patients have early stage disease. MRI is useful for evaluation of masses that are difficult to visualize with ultrasound; nevertheless, gadolinium has been found to cross the placenta and to stimulate malformations in animal models. Hence, its use during pregnancy is contraindicated in the first trimester of pregnancy and improved MRI techniques are warranted.
Recommended reference: Boussios S, et al. A review on pregnancy complicated by ovarian epithelial and non-epithelial malignant tumors: Diagnostic and therapeutic perspectives. J Adv Res. 2018;12:1-9.
Response: Thank you for your comments. We will add your comments and recommended reference.
[2] “Discussion”, Page 8 of 11, Lines 226-228:
“Thus, we think that increasing proportion of non-SCC cervical cancer might influence the incidence differences of other pathologic findings because it provides worse prognosis than SCC cervical cancer”.
At that point, the authors should clarify that the difference in survival outcomes of squamous cell carcinomas and adenocarcinomas of the uterine cervix is still controversial. It has not been elucidated whether patients with cervical adenocarcinomas have a higher incidence of pelvic lymph node involvement compared to those with squamous cell carcinomas. Probably, the worse outcome of patients with adenocarcinomas is due to ineffective adjuvant treatment, rather than a higher incidence of lymph node involvement as compared to squamous cell carcinomas.
Response: Thank you for your comments. We will rephrase the statement on which you comment.
[3] General comment:
A workflow diagram for the study would be of benefit for the readers.
Response: Thank you for your comment. We will add it in the manuscript.